# A randomised controlled feasibility trial of a BabyWASH household playspace: The CAMPI study

Sophie Budge[1], Paul Hutchings[2]*, Alison Parker[1], Sean Tyrrel[1], Sam Norton[3], Camila Garbutt[4], Fitsume Woldemedhin[5], Mohammed Yasin Jemal[5], Mathewos Moges[6], Siraj Hussen[6], Hunachew Beyene[6]

1 Cranfield Water Science Institute, Cranfield University, Cranfield, United Kingdom, 2 Faculty of Engineering and Physical Sciences, University of Leeds, Leeds, United Kingdom, 3 Psychology Department, Institute of Psychiatry Psychology & Neuroscience, King's College London, London, United Kingdom, 4 PIN UK, London, United Kingdom, 5 PIN, Hawassa, Ethiopia, 6 Hawassa University College of Medicine and Health Sciences, Hawassa, Ethiopia

☯ These authors contributed equally to this work.
* p.hutchings@leeds.ac.uk

**Data Availability Statement:** All underlying data, including study protocol and consent forms, are

## Abstract

### Background

Water, sanitation and hygiene (WASH) interventions should support infant growth but trial results are inconsistent. Frequently, interventions do not consider behaviours or transmission pathways specific to age. A household playspace (HPS) is one intervention component which may block faecal-oral transmission. This study was a two-armed, parallel-group, randomised, controlled feasibility trial of a HPS in rural Ethiopia. It aimed to recommend proceeding to a definitive trial. Secondary outcomes included effects on infant health, injury prevention and women's time.

### Methods

November 2019−January 2020 106 households were identified and assessed for eligibility. Recruited households (N = 100) were randomised (blinded prior to the trial start) to intervention or control (both n = 50). Outcomes included recruitment, attrition, adherence, and acceptability. Data were collected at baseline, two and four weeks.

### Findings

Recruitment met *a priori* criteria (≥80%). There was no loss to follow-up, and no non-use, meeting adherence criteria (both ≤10%). Further, 48.0% (95% CI 33.7−62.6; n = 24) of households appropriately used and 56.0% (41.3−70.0; n = 28) cleaned the HPS over four weeks, partly meeting adherence criteria (≥50%). For acceptability, 41.0% (31.3−51.3; n = 41) of infants were in the HPS during random visits, failing criteria (≥50%). Further, the proportion of HPS use decreased during some activities, failing criteria (no decrease in use). A modified Barrier Analysis described good acceptability and multiple secondary benefits, including on women's time burden and infant injury prevention.

freely available at: https://doi.org/10.17862/cranfield.rd.14877831.v1.

**Funding:** SB is jointly funded as a research student by both Cranfield University and People In Need, who received funding from the Czech Development Agency for the project. No other external funds supported this work. The funders had no role in study design, data collection and analysis, decision to publish, or preparation of the manuscript.

**Competing interests:** The authors have declared that no competing interests exist.

## Interpretation

Despite failing some *a priori* criteria, the trial demonstrated mixed adherence and good acceptability among intervention households. A definitive trial to determine efficacy is warranted if recommended adjustments are made.

## Funding

People In Need; Czech Development Agency.

## Trial registration

RIDIE-ID-5de0b6938afb8.

## Author summary

This research tested a new way to protect infants and young children from infections that are caused by pathogens in human and animal faeces. It tested the feasibility of using a household playspace to reduce infection by creating a hygienic environment for children to play-in in rural Ethiopia. The results show that the household playspace was well accepted, used regularly and cleaned well by participants in the study. The study also suggests a potential positive impact in reducing diarrhoea. Based on these results, we suggest that a larger scale trial be conducted to conclusively assess whether a household playspace can protect young children and infants from infection in rural Ethiopia or similar contexts.

## Introduction

Final height in adults results from both genetic and environmental factors which support linear growth in childhood [1]. Conversely, adverse influences which begin *in utero* and continue through puberty can lead to growth failure [1]. This includes the cyclical relationship between infection and nutrition. Symptomatic infection is common during early years in low-income countries, and repeated diarrhoea impairs growth, weight gain and long-term cognitive development [2]. Moreover, enteric infections which are asymptomatic but which result in subclinical enteropathy[3] are also associated with growth shortfalls [4,5]–suggesting infection affects development without overt outcomes like diarrhoea. Population-level nutrition and hygiene status are thus critical for proper growth, but are not sufficient alone: where there are widespread infection and inflammation, the effect of nutrition on growth is seriously compromised [1]. Indeed, the modest effects on growth in nutrition interventions suggests that a combination of recurrent infections, chronic inflammation, and gut enteropathy limit the effects of nutrition [6]. Thus randomised controlled trials (RCT) are testing water, sanitation and hygiene (WASH) interventions alongside supplementary nutrition to improve infant health.

Despite substantial evidence suggesting safe WASH contributes to good child health in terms of preventing malnutrition and morbidities from infectious diseases[7], RCTs testing improved household WASH (with or without supplementary nutrition) have shown variable, mostly insignificant, effects [8–11]. Whilst it is improbable that interventions at the coverage in these trials will alleviate growth failure, results have prompted discourse on what is necessary. The concept 'Transformative WASH'[12] highlights the necessity of substantially

improving environmental hygiene amongst the poorest, whom disproportionately experience poor child health. It also recognises the significant burden of contamination from domestic animals–largely unaddressed in WASH trials or programs[13]. In rural, subsistence agriculture settings it is common for domestic animals to share living and sleeping spaces. Acting as natural reservoirs, domestic animals likely contribute substantial contamination to multiple transmission routes of zoonotic pathogens such as *Cryptosporidium*, *Campylobacter* and *E. coli* [4,5,13–15] which are associated with growth failure and gut abnormalities [16,17]. Further, a transformative approach will require that interventions (whether technical, structural, or behavioural) consider age-related behaviours and transmission pathways to prevent infant infection [18]. One such 'critical' WASH intervention component[19] is a household playspace (HPS)–an enclosed, protective play area. In rural areas, a HPS may offer some protection from infection during early growth periods by interrupting faecal-oral transmission from ingested soil and faeces[20,21]and contaminated floors during infantplay [22].

Available evidence on the health and non-health benefits of a HPS or playmat has been previously reviewed [23]. This included preventing the ingestion of faeces and contaminated matter (soil, other objects) by the infant and protection from injury. Further, formative data during the participatory design and build of the HPS prototype suggested caregivers liked it and were glad to use it during daily routines [23]. However, there remains a need to assess how long a HPS would be used throughout the day and appropriately maintained and cleaned. Data on infant health outcomes would provide insight into the potential for a HPS to reduce infection from within the home. Moreover, WASH interventions deliver both health and non-health outcomes, all of which contribute to household wellbeing. Thus broader benefits of a HPS, including on women' time and child socioemotional development, also require exploration through a definitive RCT.

### Aims

The Campylobacter-Associated Malnutrition Playspace Intervention (CAMPI) trial was a randomised, controlled feasibility trial to establish the feasibility of a definitive RCT of a HPS in rural Ethiopia. The HPS design (S1 Figure), is described elsewhere [23], underpinned by previous formative research [24–26]. The primary aim of the trial was to establish the feasibility of a future definitive RCT to evaluate the efficacy of a HPS. This involved evaluating the HPS through measures of recruitment, attrition, adherence, and acceptability, and as efficacy methods within a RCT. It also involved evaluating the appropriateness of the study design for recommended adjustments to the intervention and design for future trials.

As formal hypothesis testing for effectiveness is not recommended in feasibility studies, the trial did not aim to determine the effect of the HPS on health outcomes, and was not powered for this. However, further evidence was required towards the infection-exposure hypothesis as well as effects on broader outcomes. Thus secondary outcomes aimed to:

1. Confirm the prevalence of *Campylobacter* infection in the study population

2. Describe effects of the HPS on *Campylobacter* infection and diarrhoea

3. Describe secondary effects, including on women's use of time, childcare, or injury prevention.

### Methods

This feasibility trial was designed by Cranfield University alongside People in Need (PIN) and Hawassa University and conducted in the Southern Nations, Nationalities and Peoples' region

(SNNPR), Ethiopia. It was a two-armed, parallel-group, randomised controlled feasibility trial with equal group allocation. The Consolidated Standards of Reporting Trials (CONSORT) 2010 statement with extension to pilot trials was followed during study design and reporting (S1 PRISMA Checklist).

## Ethics statement

This study is registered with the Registry for International Development Impact Evaluations (RIDIE), (RIDIE-STUDY-ID-5de0b6938afb8). The study was approved by Cranfield University Research Ethics Committee (CURES/9357/2019) and Hawassa University College of Medicine and Health Sciences Institutional Review Board (IRB/010/12). Upon recruitment, PIN staff and HEWs discussed the study with primary caregivers who understood data were anonymous. Informed consent and assent on behalf of infant participants was obtained, or thumbprints taken. Surveys were translated to Amharic by PIN staff and administered verbally in Amharic or Sidamo. Various checks throughout the trial assessed HPS safety and monitored for adverse events. This included regular survey checkpoints (data concerns from households and HPS safety and visual inspection of HPS for unsafe use or assembly), and the distribution of feedback response mechanism cards to contact PIN staff. Infants with moderate or severe acute malnutrition measured by MUAC were advised to contact their local health post, which was followed up by a government Health Extension Work (HEW) HEW.

## Randomisation and masking

As a feasibility trial, a sample size calculation based on power was not performed. A target of 100 households was deemed sufficient to inform researchers about practicalities of running the trial and for sufficient precision to estimate rates of recruitment, retention, and trial outcomes. Specifically, a sample size of 100 was deemed sufficient since to achieve a maximum standard error of ±0.05 for a proportion, which ensures a 95% confidence interval in the estimated results with a maximum width of +/-10%. This was deemed achievable with the resources available and is in line with recommendations for feasibility studies where the parameter of interest is a proportion.

Eligible households were identified, contacted and enrolled into the trial November 2019–January 2020. Four kebeles (a neighbourhood or small administrative unit; two intervention, two control) were chosen from a woreda (zonal subdivision) representative of rural livelihoods across the region, without geographical overlap. Alongside government HEW, PIN team members produced a blinded sampling frame from kebeles of all households fulfilling eligibility criteria. Households were sequentially numbered and using statistical software, 25 households were randomly drawn from each frame for a total sample of 100 (50 intervention, 50 control). Inclusion (eligibility) criteria were: 1. Subsistence agriculture households raising domestic animals, within PIN intervention scope; 2. With an infant aged 8–16 months (10–18 months at trial commencement); 3. Not participating in other PIN projects. Exclusion criteria: 1. Outside 10–18 month range at trial start; 2. Participating in other PIN projects; 3. Infant was pre-term, low birth weight, or had other birth complications. PIN staff and HEWs approached households with the study information and participants were given time to make an informed decision. Households were then revisited, eligibility was re-verified, and if households were willing, consent was gained. Households were blinded to their status in the trial until after baseline data collection. Fig 1 describes trial enrollment and numbers.

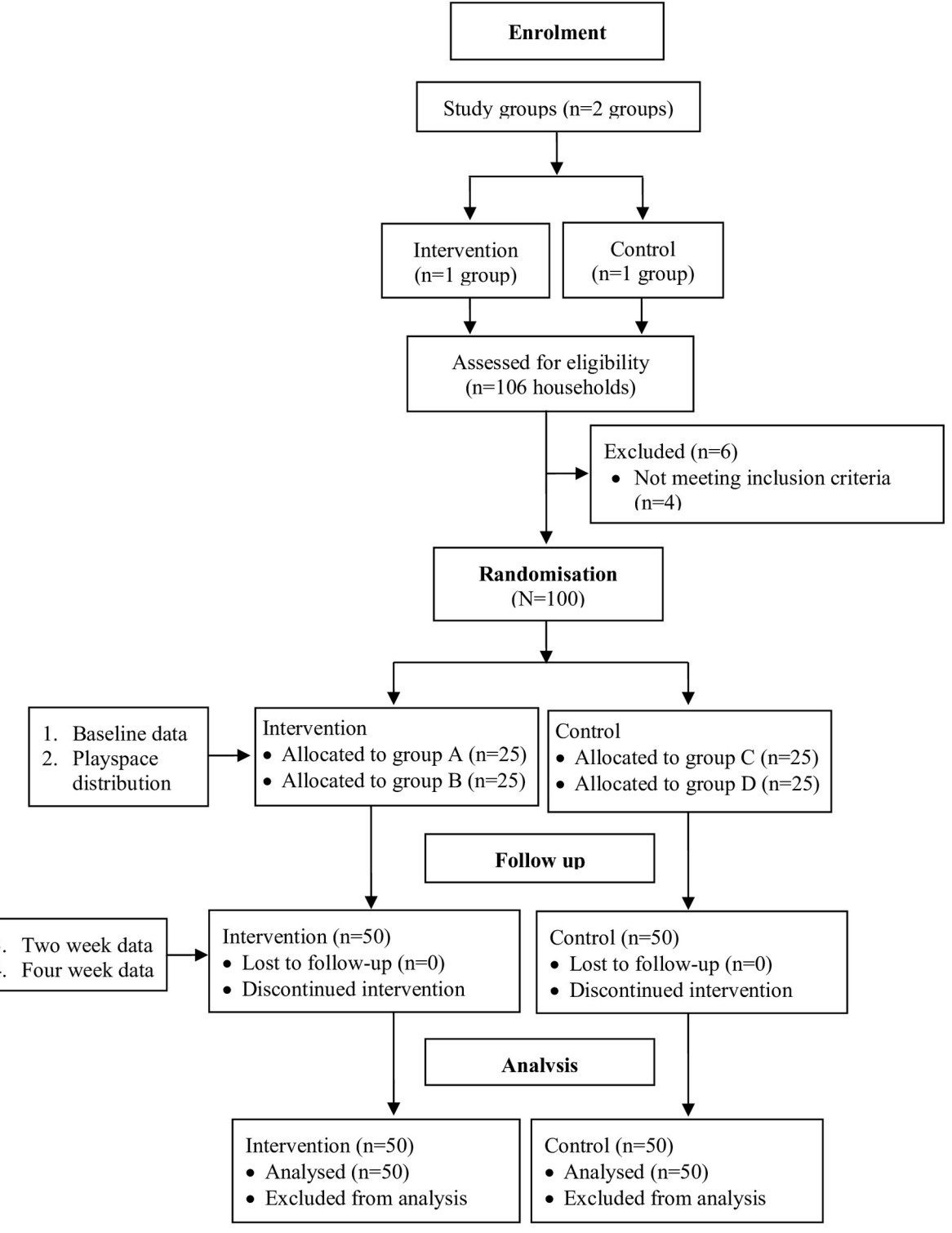

**Fig 1. Modified version of CONSORT 2010 flow diagram of participants in the CAMPI feasibility trial.**

## Study intervention

The trial was conducted in Sidama zone, SNNPR, Ethiopia between January–March 2020. Two field teams managed intervention and control kebeles. After baseline data collection, caregivers from intervention households were called to the kebele health post for a 'sensitisation' day. PIN field team, HEWs and data collectors formally discussed the study rationale, caregiver beliefs around infant faecal-oral transmission and health outcomes, transmission routes and how a HPS might interrupt these to improve infant health. Correct HPS use, maintenance and cleaning was detailed. Caregivers watched and practiced HPS assembly and discussed potential safety issues. Use was discussed in relation to daily routines and activities and caregivers agreed to use it when possible. Households agreed to clean the HPS at least every other day (and always after defecation or urination) with both soap and water. Playspaces were distributed with safety instructions printed in both Sidamo and Amharic with illustrations. HEWs visited intervention households in the following days to ensure correct HPS assembly. The control group received a HPS upon study completion.

## Participant data

**Survey and anthropometry.**   Households were visited at baseline and at two and four weeks. The primary caregiver present was interviewed, usually the mother. Baseline data included a previously validated survey[25,26] on WASH facilities and use (latrine type; presence and use, presence and availability of soap and water for handwashing; availability of water within the home; water source; person responsible for collecting water; safe storage of water) and animal husbandry practices. Food hygiene, breastfeeding, and diarrhoea incidence were also assessed and again at two and four weeks. Trained data collectors took weight, height and mid-upper arm circumference (MUAC) following standard procedure[27] with a digital mother-child smart scale (Ultratec), a foldable infantometer to 5 mm accuracy (seca 210) and standard MUAC tape to 1 mm accuracy, respectively. Seven-day diarrhoea prevalence was by caregiver report.

**Laboratory confirmation.**   Collection and processing of infant faecal samples followed a validated methodology [26]. Briefly, a day prior to household visits data collectors distributed sterile sample collection bags (Whirl-Pak, Sigma-Aldrich, UK) labelled with study time point and study ID and demonstrated sterile sample collection. Faecal samples were collected the morning of deposition and transported on ice within six hours to the laboratory at Hawassa University College of Medicine and Life Sciences. That same day, samples were processed for the isolation of presumptive *Campylobacter* spp. Microbial culture was performed by trained laboratory staff using CHROMagar selective media and appropriate microaerophilic conditions. Samples were processed for all 100 households at each of the three study time points (N = 300).

## Implementation outcomes

**Evaluation of trial outcomes and proceeding with future definitive trial.**   Among the intervention group, surveys at two and four weeks assessed feasibility outcomes: Recruitment (number of households contacted who consented); Attrition (the proportion of participants lost to follow-up at the trial end); Adherence (proportion of HPS non-use, as well as Appropriate use/maintenance and cleaning), and Acceptability (random observation of HPS use and change in incidence [proportion] of use from two to four weeks). A modified Barrier Analysis at four weeks provided further insight into acceptability. As these outcomes were the main measures to determine whether to proceed to a definitive trial, *a priori* threshold criteria were established as follows: 1. Recruitment: the proportion of contacted households participating in

the trial would be ≥80%; 2. Attrition: the level at the trial end would be ≤10%; 3. Adherence: the proportion of non-use of HPS would be ≤10% at both time points and over the trial; 4. Adherence: the proportion of correct HPS use and cleaning would be ≥50% at both time points and over the trial; 5. Acceptability: the proportion of infants in the HPS at random check would be ≥50% at both time points and over the trial, and 6. Acceptability: reported incidence of HPS use during daily activities (as a proportion) would not decrease from two to four weeks. Outcomes would also indicate appropriateness of an RCT and provide recommendations for adjusting the intervention design.

**Statistical analysis.** Data were managed in Excel and analysed in SPSS (v26.0, IBM). Descriptive statistics summarised survey data and health outcomes. Trial outcomes are displayed with estimated 95% confidence intervals (CI), calculated using the Wald method for proportions. The adherence outcome included 'Appropriate use' and 'Appropriate cleaning', created as composite binary outcome variables (described in table footnotes) and described across study time points. Adherence as 'HPS non-use' was described as reported non-use after baseline. Acceptability as 'Infant in playspace upon arrival' was calculated for both visits. Acceptability as change in HPS use was calculated from reported HPS use during reported daily activities over two and four weeks and the difference in proportions. A Generalised Estimating Equation (GEE) was used as a semi-parametric model, using a robust variance estimator and an unstructured working correlation matrix. A binary logistic GEE estimated factors associated with 'Appropriate use' and 'Appropriate cleaning' at two and four weeks. Models were initially run separately: however, the merged composite variable of 'Appropriate use and cleaning' showed no difference in parameter estimates between models and is presented. Prespecified variables included infant sex and age; maternal age; maternal education; number in household; number of children; household owns soap; safe water storage; animal husbandry practices; water availability, and mother collects water. Results are expressed as populated averaged odds ratios (ORs) with estimated 95% CI. As this is a feasibility study, in line with the CONSORT extension for feasibility studies, we do not present p-values.

Acceptability was further assessed through a modified Barrier Analysis which explored determinants of use among all participants. Methods and analysis are described in detail in supplementary information (S1 Text). Derivation of themes was data-driven, where codes resulted from the analysed data as they related to each determinant (See Table D in S1 Data). Coded themes are discussed as either barriers or enablers to the implantation of, and improving outcomes during, a definitive trial. For secondary health outcomes, anthropometric z-scores were calculated (WHO Anthro v3.2.2) and categorised into stunting and wasting using standard cut-off values [28]. Samples positive for presumptive *Campylobacter* spp., colonies were counted using OpenCFU. Change in diarrhoeal and *Campylobacter* prevalence between study groups was estimated using a GEE intercept-only model with OR and 95% CI.

## Results

### Baseline characteristics

Household demographic characteristics are described in Table 1 for both study groups and as a whole. Characteristics were largely balanced across groups. Average infant age was 10.8 months (median 10.0; range 7–18). Average length-for-age (LAZ) and weight-for-length (WLZ) at baseline did not vary substantially across intervention and control groups at -1.00 and -0.96 (LAZ) and -0.49 and -0.46 (WLZ) respectively. Stunting and wasting affected 33.0% (n = 33) and 13.0% (n = 13) of all infants respectively with some severe acute malnutrition (11.0%, n = 11). Mothers were mostly aged 18–25 (50.0%, n = 25; 62.0%, n = 31, respectively) and educated to second grade (44.0%, n = 22; 52.0%, n = 26, respectively). Whilst most

**Table 1.** Household demographic characteristics, and water, sanitation and hygiene, animal husbandry and nutrition indicators across study groups and as a total at baseline (N = 100).

| | Intervention (n = 50) | | Control (n = 50) | | Total (N = 100) | |
|---|---|---|---|---|---|---|
| | n | % | n | % | n | % |
| **Demographics** | | | | | | |
| Infant sex: Male | 28 | 56.0 | 24 | 48.0 | 52 | 52.0 |
| Average infant age (months) | 10.1 | | 11.6 | | 10.8 | |
| Respondent: Mother | 45 | 90.0 | 48 | 96.0 | 93 | 93.0 |
| Maternal age: <18 | 1 | 2.0 | 0 | 0.0 | 1 | 1.0 |
| 18–25 | 25 | 50.0 | 31 | 62.0 | 56 | 56.0 |
| 26–35 | 21 | 42.0 | 18 | 36.0 | 39 | 39.0 |
| 36–45 | 3 | 6.0 | 0 | 0.0 | 3 | 3.0 |
| >45 | 0 | 0.0 | 1 | 2.0 | 1 | 1.0 |
| Maternal education: Cannot read/write | 12 | 24.0 | 6 | 12.0 | 18 | 18.0 |
| First grade | 13 | 26.0 | 15 | 30.0 | 28 | 28.0 |
| Second grade | 22 | 44.0 | 26 | 52.0 | 48 | 48.0 |
| Secondary and above | 3 | 6.0 | 3 | 6.0 | 6 | 6.0 |
| Number in household: 1–3 | 5 | 10.0 | 9 | 18.0 | 14 | 14.0 |
| 4–6 | 34 | 68.0 | 33 | 66.0 | 67 | 67.0 |
| 7+ | 11 | 22.0 | 8 | 16.0 | 19 | 19.0 |
| Number of children: 1–2 | 22 | 44.0 | 23 | 46.0 | 45 | 45.0 |
| 3–4 | 18 | 36.0 | 23 | 46.0 | 41 | 41.0 |
| 5–6 | 9 | 18.0 | 4 | 8.0 | 13 | 13.0 |
| 7+ | 1 | 2.0 | 0 | 0.0 | 1 | 1.0 |
| Number of children ≤5: 1 | 35 | 70.0 | 36 | 72.0 | 71 | 71.0 |
| 2 | 12 | 24.0 | 13 | 26.0 | 25 | 25.0 |
| 3 | 3 | 6.0 | 1 | 2.0 | 4 | 4.0 |
| Main income: Farming/livestock | 48 | 96.0 | 49 | 98.0 | 97 | 97.0 |
| Trade | 17 | 34.0 | 19 | 38.0 | 36 | 36.0 |
| Employee | 1 | 2.0 | 0 | 0.0 | 1 | 1.0 |
| Household has formal means of saving | 9 | 18.0 | 8 | 16.0 | 17 | 17.0 |
| House material: Wood and mud | 38 | 76.0 | 47 | 94.0 | 85 | 85.0 |
| Wood and grass | 10 | 20.0 | 1 | 2.0 | 11 | 11.0 |
| Concrete | 2 | 4.0 | 2 | 4.0 | 4 | 4.0 |
| Floor material: Concrete / cement | 16 | 32.0 | 9 | 18.0 | 25 | 25.0 |
| Mud / soil | 34 | 68.0 | 41 | 82.0 | 75 | 75.0 |
| | **Intervention** (n = 50) | | **Control** (n = 50) | | **Total** (N = 100) | |
| | n | % | n | % | n | % |
| **WASH indicators** | | | | | | |
| Latrine type: Defecate in open | 8 | 16.0 | 11 | 5.0 | 19 | 19.0 |
| Share neighbour's | 6 | 12.0 | 8 | 16.0 | 14 | 14.0 |
| Pit latrine without slab | 11 | 22.0 | 5 | 10.0 | 16 | 16.0 |
| Pit latrine with slab | 25 | 50.0 | 26 | 52.0 | 51 | 51.0 |
| Water source: Piped water / public tap | 50 | 50.0 | 50 | 50.0 | 100 | 100.0 |
| Who collects water?: Mother | 39 | 78.0 | 39 | 78.0 | 78 | 100.0 |
| Father | 8 | 16.0 | 3 | 6.0 | 11 | 11.0 |
| A grandparent | 0 | 0.0 | 1 | 2.0 | 1 | 1.0 |
| Female child (≤15) | 11 | 22.0 | 10 | 20.0 | 21 | 21.0 |

(*Continued*)

**Table 1.** (Continued)

| | n | % | n | % | n | % |
|---|---|---|---|---|---|---|
| Male child (≤15) | 3 | 6.0 | 7 | 14.0 | 10 | 10.0 |
| Labourer | 1 | 2.0 | 11 | 22.0 | 12 | 12.0 |
| Water available inside the home | 43 | 86.0 | 46 | 92.0 | 89 | 89.0 |
| Household safely stores water* | 12 | 24.0 | 14 | 28.0 | 26 | 26.0 |
| Household owns soap | 34 | 68.0 | 39 | 78.0 | 73 | 100.0 |
| **Animal husbandry** | | | | | | |
| Number of cattle: 1–3 | 31 | 62.0 | 25 | 25.0 | 56 | 56.0 |
| 4–6 | 9 | 18.0 | 11 | 22.0 | 20 | 20.0 |
| 7+ | 0 | 0.0 | 1 | 2.0 | 1 | 1.0 |
| Number of goats: 1–3 | 8 | 16.0 | 8 | 16.0 | 16 | 16.0 |
| 4–6 | 2 | 4.0 | 1 | 2.0 | 3 | 3.0 |
| 7+ | 1 | 2.0 | 1 | 2.0 | 2 | 2.0 |
| Number of donkey: 1–3 | 1 | 2.0 | 6 | 12.0 | 7 | 7.0 |
| Number of sheep: 1–3 | 0 | 0.0 | 10 | 20.0 | 10 | 10.0 |
| Number of chickens: 1–3 | 11 | 22.0 | 16 | 32.0 | 27 | 27.0 |
| 4–6 | 15 | 30.0 | 16 | 32.0 | 31 | 31.0 |
| 7+ | 18 | 36.0 | 7 | 14.0 | 25 | 25.0 |
| Animal dwelling during the day | | | | | | |
| Outside, enclosed in a pen | 1 | 2.0 | 0 | 0.0 | 1 | 1.0 |
| Outside, roaming free | 49 | 98.0 | 48 | 96.0 | 97 | 97.0 |
| Inside, same room as family | 33 | 66.0 | 40 | 80.0 | 73 | 73.0 |
| Inside, separate room | 1 | 2.0 | 2 | 4.0 | 3 | 3.0 |
| Animal dwelling during the night | | | | | | |
| Outside, enclosed in a pen | 7 | 14.0 | 10 | 20.0 | 17 | 17.0 |
| Outside, roaming free | 0 | 0.0 | 0 | 0.0 | 0 | 0.0 |
| Inside, same room as family | 34 | 68.0 | 35 | 70.0 | 69 | 69.0 |
| Inside, separate room | 9 | 18.0 | 5 | 10.0 | 14 | 14.0 |

| | Intervention (n = 50) | | Control (n = 50) | | Total (N = 100) | |
|---|---|---|---|---|---|---|
| | n | % | n | % | n | % |
| **Nutrition indicators** | | | | | | |
| LAZ z-score (average) | -1.00 | | -0.96 | | -0.98 | |
| LAZ (range) | -3.04−0.80 | | -2.76−0.66 | | -3.04−0.80 | |
| WLZ z-score (average) | -0.49 | | -0.46 | | -0.47 | |
| WLZ (range) | -2.30−0.87 | | -2.41−0.75 | | -2.41−0.87 | |
| MUAC[a] (mm; average) | 138.2 | | 138.1 | | 138.2 | |
| Stunting (LAZ ≤ −2 SD) | 17 | 34.0 | 16 | 32.0 | 33 | 33.0 |
| Wasting (WLZ ≤ −2 SD) | 6 | 12.0 | 7 | 14.0 | 13 | 13.0 |
| MUAC[a]: ≥135 | 36 | 72.0 | 39 | 78.0 | 75 | 75.0 |
| 125−135 | 8 | 16.0 | 6 | 12.0 | 14 | 13.0 |
| 115−124 | 6 | 12.0 | 5 | 10.0 | 11 | 11.0 |
| ≤115 | 0 | 0.0 | 0 | 0.0 | 0 | 0.0 |

WASH, water, sanitation and hygiene; LAZ, length-for-age; WLZ, weight-for-length; MUAC, mid-upper arm circumference, where: ≥135, no risk of undernutrition; 12.5–13.5, at risk of moderate acute undernutrition; 11.5–12.4, moderate acute undernutrition; ≤11.5, severe acute undernutrition.

*Calculated as households who were marked 'Yes' to all three observation-based questions: Are water containers clean; Do the water containers have a protecting cover; Does the container have a tap or narrow mouth for drawing the water.

households had a pit latrine with a slab (51.0%, n = 51), open defecation was still common (19.0%, n = 19). Seventy-eight percent of women bore the duty of collecting water which for all households came from a public tap. Only 26.0% of households safely stored their water. Cattle and chickens were the most frequent domestic animal, and husbandry practices indicated animals frequently shared living spaces during the day and night, with infrequent use of pens.

## Trial outcomes

For ease of assessment, study outcomes are described together in Table 2 and individually in sections below. Given that the analysis was exploratory in this feasibility trial and results were preliminary, the 95% CI is expressed without p values.

## Recruitment and attrition

Rates for recruitment and attrition are shown in Table 2. One hundred households were recruited from four kebeles. To achieve this, 106 households were assessed for eligibility; four households were then excluded for not meeting infant age criteria at the study start and a further two did not consent to participate (Fig 1). Thus a recruitment rate of 94.3% (95% CI 88.1 −97.9) met *a priori* criteria of ≥80%. All households completed the trial assessments at four weeks and there was no loss to follow-up (0.0%; 95% CI 0.0−3.6), meeting criteria for attrition (≤10% at trial end).

## Adherence

Adherence was first described as the proportion of HPS non-use at both time points and over the study period (Table 2). No households reported not using the HPS at either time point or over the study duration (0.0%, 95% CI 0.0−0.71), meeting *a priori* criteria ≤10%. Second, adherence was described through 'Appropriate use' and 'Appropriate cleaning' and combined, across the study time points and throughout the trial (Table 3). Appropriate use included maintenance, as described in the table footnotes alongside variable components (also in Table A in S1 Data). When considering behaviours and time points separately, Appropriate use and Appropriate cleaning were consistently above the *a priori* threshold of 50%. However when assessing throughout, findings are mixed. Appropriate use did not meet the threshold (48.0%) whilst cleaning did (56.0%) and only 26.0% of households appropriately used and cleaned the HPS throughout the trial. Variables associated with adherence outcomes across

**Table 2. Outcomes for the CAMPI trial to determine progression to a future definitive RCT, at two and four weeks and across the trial duration.**

| Quantitative trial outcomes | | | | | | | | | | |
|---|---|---|---|---|---|---|---|---|---|---|
| Outcome | Definition / Indicator | *A priori* criteria | Proportion (N = 50) | | | | | | | |
| | | | Baseline | | Two weeks | | Four weeks | | Study duration | |
| | | | % | 95% CI | % | 95% CI | % | 95% CI | % | 95% CI |
| Recruitment | Proportion of contacted houses who consented | ≥80% | 94.3 | 88.1−97.9 | - | - | - | - | - | - |
| Attrition | Loss to follow-up | ≤10% | - | - | - | - | - | - | 0.0 | 0.0−3.6 |
| Adherence | Non-use of HPS | ≤10% | - | - | 0.0 | 0.0−0.07 | 0.0 | 0.0−0.07 | 0.0 | 0.0−0.07 |
| | Appropriate use | ≥50% | - | - | 70.0 | 55.5−82.1 | 64.0 | 49.2−77.1 | 48.0 | 33.7−62.6 |
| | Appropriate cleaning | ≥50% | - | - | 72.0 | 57.5−83.8 | 70.0 | 55.4−82.1 | 56.0 | 41.3−70.0 |
| | Appropriate use and cleaning | ≥50% | - | - | 52.0 | 37.4−66.3 | 48.0 | 33.7−62.6 | 26.0 | 14.6−40.3 |
| Acceptability | Infant in HPS upon arrival | ≥50% | - | - | 32.0 | 19.5−46.7 | 50.0 | 35.5−64.5 | 41.0 | 31.3−51.3 |
| | Proportion of HPS use during daily activities | No decrease | - | - | - | - | - | - | Decrease during certain activities | |

**Table 3. Adherence: Appropriate playspace use and cleaning across study time points.**

| Adherence: Appropriate HPS use and cleaning (N = 50) | | | | | | | | | |
|---|---|---|---|---|---|---|---|---|---|
| | Two weeks | | | Four weeks | | | Both time points[β] | | |
| | n | % | 95% CI | n | % | 95% CI | n | % | 95% CI |
| Appropriate use[*] | 35 | 70.0 | 55.5–82.1 | 32 | 64.0 | 49.2–77.1 | 24 | 48.0 | 33.7–62.6 |
| Appropriate cleaning[**] | 36 | 72.0 | 57.5–83.8 | 35 | 70.0 | 55.4–82.1 | 28 | 56.0 | 41.3–70.0 |
| Appropriate use and cleaning[α] | 26 | 52.0 | 37.4–66.3 | 24 | 48.0 | 33.7–62.6 | 13 | 26.0 | 14.6–40.3 |

HPS, household playspace; CI, confidence interval.

[*]Created from the variables: Playspace is assembled correctly (observed), yes; Any changes/modifications to playspace (observed), no, or yes, modifications are safe; Others share playspace (reported), no; Animals in playspace (observed and reported), no; Caregiver leaves infant in playspace when leaving house (reported), no or yes IF; infant is watched by other adult (father, grandparent or child ≥18).

[**]Created from the variables: Frequency of cleaning the playspace (reported), every day, every other day; Cleaning materials used (reported), water and soap; Mattress visibly dirty (observed), no; Urine or faeces on mattress (human or animal; reported), no.

[α]The sum of households who achieved 'Yes' for all criteria for both use and cleaning.

[β]The sum of households who achieved 'Yes' for all criteria across indicators at both two and four weeks.

the two time points were assessed using a binary logistic GEE model (Table 4). Results display the 95% CI for the effect size and odds ratio. The only variable to significantly predict Appropriate use or cleaning was 'Mother collects water alone', where an inverse relationship showed a reduced odds of 72.0% (0.28; 95% CI 0.12–0.66).

## Acceptability

**Infant in playspace upon arrival, change in playspace use.** The first measure noted if the infant was in the HPS during a random visit (Table 2). This increased from 32.0% (95% CI 19.5–46.7, n = 16) at two weeks to 50% (95% CI 35.5–64.5, n = 25) at four weeks, meeting *a priori* criteria of 50% at this point: however throughout the trial did not reach the threshold

**Table 4. Adherence: A Generalised Estimation Equation estimating the effect of parameters on the trial outcome of adherence 'Appropriate use and cleaning' across study time points.**

| Adherence: Appropriate use and cleaning Generalised Estimating Equation (N = 50) | | | | | | | |
|---|---|---|---|---|---|---|---|
| Variable | | | 95% Wald Confidence Interval | | | 95% Wald Confidence Interval for Exp(B) | |
| | B | Std. Error | Lower | Upper | Odds Ratio | Lower | Upper |
| (Intercept) | -0.07 | 1.06 | -2.14 | 2.01 | 0.94 | 0.12 | 7.49 |
| Infant sex = Male | 0.42 | 0.50 | -0.57 | 1.40 | 1.52 | 0.57 | 4.06 |
| Maternal age = ≤25 | -0.65 | 0.70 | -2.02 | 0.71 | 0.52 | 0.13 | 2.04 |
| Maternal education = Illiterate | -0.56 | 0.61 | -1.75 | 0.63 | 0.57 | 0.17 | 1.88 |
| Number in household = 1–3 | -0.0 | 0.78 | -2.52 | 0.52 | 0.37 | 0.08 | 1.68 |
| Number of children = 1–2 | 0.45 | 0.59 | -0.71 | 1.60 | 1.56 | 0.49 | 4.96 |
| Household owns soap = 1 | 0.63 | 0.53 | -0.41 | 1.67 | 1.87 | 0.66 | 5.31 |
| Water is safely stored = Yes | -0.11 | 0.54 | -1.17 | 0.95 | 0.90 | 0.31 | 2.59 |
| Animals inside day = Yes | -0.22 | 0.53 | -1.26 | 0.81 | 0.89 | 0.28 | 2.25 |
| Animals inside night = Yes | -0.38 | 0.69 | -1.73 | 0.97 | 0.68 | 0.18 | 2.63 |
| Water available = Yes | -0.55 | 0.75 | -2.03 | 0.93 | 0.58 | 0.13 | 2.54 |
| Mother collects water alone = Yes | -1.28 | 0.44 | -2.15 | -0.42 | 0.28 | 0.12 | 0.66 |
| Infant age (scale) | 0.12 | 0.10 | -0.07 | 0.31 | 1.13 | 0.93 | 1.37 |

**Table 5. Acceptability: Reported playspace use in the past 24 hours during different daily activities, at two and four weeks, and the change across time points.**

Acceptability: Reported HPS use during specified household activities by 24-hour recall and change in proportion of use across study time points (N = 50)

| Reported daily activity | Total reported activity* | Reported HPS use | Proportion of use** | Total reported activity* | Reported HPS use | Proportion of use ** | Change in use | Change in proportion of use$^{\alpha}$ |
|---|---|---|---|---|---|---|---|---|
| | Two weeks | | | Four weeks | | | Across time points | |
| | n | n | % | n | n | % | n | % |
| **Prepared / ate a meal** | **150** | **139** | **92.7** | **172** | **139** | **80.8** | **0** | **-11.9** |
| Prepared breakfast | 45 | 40 | 88.9 | 47 | 42 | 89.4 | 2 | +0.5 |
| Prepared lunch/ snacks | 56 | 54 | 96.4 | 51 | 48 | 94.1 | -6 | -2.3 |
| Prepared dinner | 49 | 45 | 91.8 | 49 | 43 | 87.8 | -2 | -4.1 |
| Ate a meal | 0 | 0 | 0.0 | 25 | 6 | 0.0 | 6 | 0.0 |
| **Prepared coffee** | **75** | **69** | **92.0** | **95** | **85** | **89.5** | **16** | **-2.5** |
| **Duties within the home** | **57** | **46** | **80.7** | **73** | **62** | **84.9** | **16** | **+4.2** |
| Cleaned the house | 47 | 40 | 85.1 | 55 | 47 | 85.5 | 7 | +0.3 |
| Washed clothes | 10 | 6 | 60.0 | 18 | 15 | 83.3 | 9 | +23.3 |
| **Duties outside of the home** | **43** | **35** | **81.4** | **49** | **43** | **87.8** | **8** | **+6.4** |
| Fetched water | 41 | 40 | 97.6 | 48 | 44 | 91.7 | 4 | -5.9 |
| Prepared enset | 30 | 25 | 83.3 | 20 | 19 | 95.0 | -6 | +11.7 |
| Chopped wood | 7 | 6 | 85.7 | 8 | 6 | 75.0 | 0 | -10.7 |
| Farmed / maintained shop | 6 | 4 | 66.7 | 21 | 18 | 85.7 | 14 | +19.0 |
| **Visits outside home** | **23** | **17** | **73.9** | **32** | **15** | **46.9** | **-2** | **-27.0** |
| Went to church / meeting | 4 | 1 | 25.0 | 4 | 0 | 0.0 | -1 | -25.0 |
| Went to market | 15 | 14 | 93.3 | 16 | 12 | 75.0 | -2 | -18.3 |
| Visited neighbours/ other | 4 | 2 | 50.0 | 12 | 3 | 25.0 | 1 | -25.0 |
| **Breastfed / fed baby** | **26** | **15** | **57.7** | **30** | **20** | **66.7** | **5** | **+9.0** |
| **Slept / rested** | **23** | **5** | **21.7** | **18** | **3** | **16.7** | **-2** | **-5.1** |

HPS, household playspace.

*Number represents reported incidence of that activity within the past 24 hours. Households (N = 50) were asked an open-ended question about their daily activities during the past 24 hours. Not every activity was reported by every respondent.

**Calculated as the proportion of households who reported using the HPS during that daily activity.

$^{\alpha}$Calculated as the difference between the proportions of HPS use at two and four weeks

(41.0%, 95% CI 31.3–51.3; n = 41). Second, change in incidence (as a proportion) of HPS during daily activities was assessed. Primary caregivers were asked open-ended questions to record their activities during the past 24 hours, and if they did or did not use the HPS. Results are shown in Table 5, with activities categorised. Broadly, there was no change in use throughout the trial during food preparation/eating but use increased during other activities inside the home (such as breastfeeding) and outside, such as preparing enset and farming. A full table describing activities and HPS use or non-use is in Table B in S1 Data. Lastly analysing HPS use according to the time of day suggested use was consistently highest in the mornings, although evening use increased at the trial end (Table E in S1 Data).

## Modified barrier analysis

Acceptability was further assessed through a semi-structured questionnaire as a modified Barrier Analysis. This assessed 12 categories of behavioural determinants, exploring all factors which would act as barriers or enablers during a definitive trial (S1 PRISMA Checklist). Full results are in Table D in S1 Data. The first seven determinants quantitatively assess beliefs and behaviours relating to infant health and HPS use. The further six determinants explored attitudes and beliefs through open-ended questions. Many cited advantages, both related and unrelated to infant health, indicated good acceptability of the HPS. Caregivers frequently stated the HPS helped prevent ingestion of dirt and faeces (80.0%, n = 40), 76.0% (n = 38). Further, many suggested the HPS prevented injury from several causes, including from fire, drowning and animals. Over half of caregivers (mothers) asserted that the HPS eased their workload (56.0%, n = 28), reduced time pressures (46.0%, n = 23) and allowed them to carry out their duties without distraction. Mothers reported relief that the HPS alleviated fears and worries over their infant's safety (52.0%, n = 26), and almost half believed their infant would physically grow better (42.0%, n = 21). Approval within the community was high among neighbours (96.0%, n = 48) husbands (40.0%, n = 20), and both close (66.0%, n = 33) and wider family (36.0%, n = 18). Conversely, some caregivers mentioned that neighbours (8.0%, n = 4) or friends (12.0%, n = 6) were envious as the common reason for disapproval (*'My friend who does not have one wants one too'*), or that money would have been preferable (*'My colleague says better to give the child clothes or money for me'*).

Barriers to use included the cost of cleaning materials (22.0%, n = 11)–echoed in the *Access* determinant where caregivers frequently noted the expense of soap (56.0%, n = 28) and cleaning materials, e.g. brushes (24.0%, n = 12). Importantly, having no older children to watch the infant was a barrier (32.0%, n = 16) and relates to the burden of workload on women. A lack of toys was also a barrier (32.0%, n = 16). Whilst the design appeared largely acceptable, some difficulties included fitting the rope connecting walls (38.0%, n = 19; see S1 Figure).

## Secondary outcomes: Infant health outcomes

Table 6 shows changes in reported seven-day diarrhoeal prevalence and presumptive *Campylobacter* spp. across groups and time periods. Considering change in point prevalence, seven-day diarrhoea declined more markedly within the intervention group from 19 cases (38.0%) at baseline to 5 cases (10.0%) at four weeks, versus 22 cases (44.0%) to 16 cases (32.0%) amongst controls. Considering change in prevalence from baseline, the intervention group showed a reduced odds of reported diarrhoea versus controls (OR 0.57, 95% CI 0.40–0.83). Baseline prevalence of presumptive *Campylobacter* was high, mirroring a similar prevalence at this site and others [26,29]. However from baseline, point prevalence showed no significant difference between groups or time points. Similarly the intervention group had no reduced odds of a *Campylobacter*-positive stool versus controls from baseline. Colony counts from positive samples can be viewed in Table D in S1 Data.

## Harms

No adverse events were observed from HPS use in the intervention group. No household reported any safety concerns associated with use, aside from one household who mentioned the plastic mattress became hot under the sun. HPS use did not increase the risk of any adverse infant health outcome, where the direction of effect does not show an increased risk for the intervention group.

**Table 6. Secondary health outcomes: Point prevalence across study time points and change in prevalence from baseline for seven-day diarrhoea and Campylobacter, intervention and control.**

**Secondary outcomes: Change in infant health outcomes**

**Reported seven-day diarrhoea point prevalence across study time points**

| | Baseline | | | | Two weeks | | | | Four weeks | | | |
|---|---|---|---|---|---|---|---|---|---|---|---|---|
| | Intervention (n = 50) | | Control (n = 50) | | Intervention (n = 50) | | Control (n = 50) | | Intervention (n = 50) | | Control (n = 50) | |
| | n | % | n | % | n | % | n | % | n | % | n | % |
| No diarrhoea | 31 | 62.0 | 28 | 56.0 | 44 | 88.0 | 35 | 70.0 | 45 | 90.0 | 34 | 68.0 |
| Diarrhoea | 19 | 38.0 | 22 | 44.0 | 6 | 12.0 | 15 | 30.0 | 5 | 10.0 | 16 | 32.0 |

**Presumptive *Campylobacter* point prevalence across study time points**

| | n | % | n | % | n | % | n | % | n | % | n | % |
|---|---|---|---|---|---|---|---|---|---|---|---|---|
| No infection | 23 | 46.0 | 24 | 48.0 | 33 | 66.0 | 32 | 64.0 | 36 | 72.0 | 36 | 72.0 |
| Infection | 27 | 54.0 | 26 | 52.0 | 17 | 34.0 | 18 | 36.0 | 14 | 28.0 | 14 | 28.0 |

**Change in reported seven-day diarrhoeal prevalence after baseline[*]**

| | Intervention (n = 50) | | Control (n = 50) | |
|---|---|---|---|---|
| | n | % | n | % |
| No diarrhoea** | 39 | 78.0 | 28 | 56.0 |
| Diarrhoea | 11 | 22.0 | 22 | 44.0 |

**Change in presumptive *Campylobacter* prevalence after baseline[β]**

| | Intervention (n = 50) | | Control (n = 50) | |
|---|---|---|---|---|
| No infection[α] | 30 | 60.0 | 28 | 56.0 |
| Any infection | 20 | 40.0 | 22 | 44.0 |

[*]OR for intervention group 0.49 (95% CI 0.33–0.75)

**No diarrhoea: No reported diarrhoea at two or four weeks, OR no reported diarrhoea from baseline; Diarrhoea: Reported diarrhoea at two or four weeks, OR reported diarrhoea from baseline.

[β]Insignificant.

[α]Negative: No suspected *Campylobacter* at two or four weeks, OR always negative; Positive: Suspected *Campylobacter* prevalence at two or four weeks, OR always positive.

## Discussion

The CAMPI trial is the first randomised, controlled feasibility trial of a HPS in rural, subsistence agriculture households in Ethiopia. Though trial outcomes did not fully reach *a priori* criteria, the trial demonstrated mixed adherence and good acceptability. On this basis, a definitive RCT for efficacy is feasible if recommended adjustments are made. Results echo two similar studies. In the SHINE trial in Zimbabwe, an imported plastic HPS and locally sourced plastic playmat were included in a WASH intervention to improve growth and anaemia. Whilst fidelity of delivery was high [10], the WASH intervention did not prevent infection [30]. However, the analysis did not estimate a magnitude of effect from the HPS specifically. In Zambia, a community-built HPS was assessed alongside a plastic model for acceptability and feasibility[31]. Reported use was similar between the two types (ranging from 10 minutes to three hours), family and community reactions suggested acceptability was high and caregiver reports suggested the community built space prevented infant ingestion of soil and animal faeces. Thus growing evidence supports wide acceptability and feasibility across different contexts and further rigorous assessment of efficacy is merited.

Addressing barriers to appropriate use and cleaning of the HPS would improve these outcomes. Data here described a broadly consistent pattern over the four weeks, albeit with a small decline (Tables 2 and 3). The modified Barrier Analysis offered reasons for diminishing use and drops in compliance, including the expense of soap and other cleaning materials. Providing these alongside the HPS would be a key consideration for any future RCT to ensure

good hygiene. Similarly, contextual WASH factors, such as water quality, availability, and unsafe storage (76.0%, n = 38 in the intervention group; Table 1) must be considered which may result in increased bacterial transmission. Similarly, the team decided not to provide toys during the trial given the potential to become vectors for indirect faecal-oral transmission [18,32].[18,32] However, this was a frequently cited barrier for mothers whose infants became bored and cried: thus providing toys or including stimulating features to the HPS is an important consideration. Alternatively caregivers may be counselled on providing (non-porous), non-hazardous toys and on regular proper cleaning. Further, during early, critical growth periods there are other important considerations including psychosocial and neurodevelopment. Opportunities for linguistic, socioemotional, and cognitive development are critical and a future RCT should consider if a HPS reduces these opportunities through interruptions to normal play, exploration, and caregiver-infant interaction–all strongly related to contextual norms and traditions.

Through random spot checks of HPS use and change in time-use, the trial showed mixed acceptability, partly meeting *a priori* criteria (Tables 2 and 5). Reported daily use increased for certain activities, suggesting an increasing ease with incorporating the HPS into daily life. However increased use during certain activities (fetching water, farming) may indicate a complacency with infant safety inside the HPS and present a risk. These increases may account for the reduction in 'Appropriate use' at week four which included if the caregiver left the infant alone whilst outside. A key finding from the modified Barrier Analysis were the secondary effects of easing work burden, time restraints, and worries about infant health and safety for many mothers. Thus in the short term, a HPS may hold many benefits including potentially improving women's empowerment through time availability and choice, reducing anxiety, and even freeing up time to spend with her infant. However any negative long-term impacts will need to balance these. This includes a lack of infant supervision, and the risk of reinforcing women's roles as sole caregivers alongside a continuing responsibility for other domestic duties. This is reinforced by the GEE model (Table 4) where when the mother bore the duty of collecting water alone, the HPS was less likely to be used or cleaned properly. In many low-income countries, women' 'triple work burden' in the productive, reproductive and social domains impedes their well-being and may reduce engagement in childcare[33]–a pattern often inherited by older female siblings. This highlights a trade-off in encouraging more active parenting alongside existing home duties, and any intervention must ensure it does not further encumber women.

The CAMPI trial was not powered to detect any differences in health outcomes between groups and results should be interpreted accordingly. Given results were preliminary, results are expressed without claims as to definite direction of effects. However secondary infant health outcomes indicated the potential efficacy of a HPS and appropriateness of these outcomes for a future RCT. Diarrhoeal prevalence from baseline reduced among the intervention group whilst presumptive *Campylobacter* did not (Table 6). Beyond the lack of adequate power, substantial methodological limitations may profoundly affect validity. These include the reliability of caregiver-reported diarrhoea and a desirability bias within intervention households; intervention households may have over-reported diarrhoea to claim for improvements in infrastructure. No further GEE analysis was performed to explore associated variables. However, aside from a potential lack of effect of the HPS on *Campylobacter* prevalence, other pathways not interrupted by the HPS likely contributed to pathogen transmission. This includes incorrectly (re-)heated foods[34]; data on this indicated unsafe practices were common (Table F in S1 Data) where across households only 28 safely prepared all meals at both time points (Table G in S1 Data). All infants were given liquids other than breastmilk, including water, possibly contaminated through unsafe storage or other pathways. *Campylobacter*

from domestic free-range poultry appears to present an infection risk to infants[5,26] and here poultry frequently shared living and sleeping areas (Table 1). However, questions remain on what, how and where infants contract *Campylobacter*, the role of domestic animals in transmission and survival time in the environment[34]. The methodology used to isolate *Campylobacter* spp. also holds limitations[26] and a definitive RCT should consider other, more sensitive techniques such as the use of ELISA or quantitative PCR.

## Progression to a definitive RCT

Progression to a full-scale trial is merited but requires some adjustments. To improve playspace adherence and acceptance, a future definitive RCT should focus on directly addressing the barriers whilst promoting the enabling factors as identified in this feasibility trial. Whilst further behavioural 'modules' and developing caregiver knowledge might have improved outcomes, it is not always practical. During the sensitisation day the HPS was introduced in a 'scalable' manner to reduce work burden among households and HEWs who are already overworked. Rather, to achieve behavioural change it is pragmatic to directly address barriers and promote enabling factors. Knowledge alone is unlikely to prevent infant faecal-oral transmission without a material element which breaks contact, and an enabling technology may drive changes in behaviour but still requires addressing factors which support or obstruct change. Factors included in the composite variable 'Appropriate use' responsible for a decline include another child sharing the HPS. Given the potential to introduce contamination, this might be addressed by a visiting HEW as a risk factor. Similarly, 'Appropriate cleaning' declined from every day/every other day to twice a week. The direction of effect and significance in the GEE model (Table 4) is an important consideration to improving this: cleaning behaviours will not change without access to soap. To improve time-use, toys (non-porous) might be provided with counselling from HEWs on regular cleaning. Factors not modifiable to counselling are important prognostic factors and might be included as strata in group randomisation in a full RCT.

Several contextual factors undoubtedly influenced this trial's operational success, including ease of recruitment and full retention. The study kebeles, within PIN outreach, may have resulted in higher acquiescence during recruitment and consent. High retention likely results from this plus a high number of data collectors for the sample. However, it is important to note that daily data collection was intense and required serious team dedication. A larger trial would likely experience higher drop out without equivalent input: a 95% CI estimate would be between 96–100% in a power calculation and 95–100% if repeated maintaining the same effort and ratio of study personnel. Over a longer time period, this is likely unsustainable. Future sample size calculations must consider these number requirements for study personnel. Furthermore, as recipients of previous WASH interventions, the intervention group likely adopted the new intervention modality earlier than might be seen in other contexts, holding implications for external validity. Good uptake may also be seen in other contexts where NGOs have a known presence and have provided multiple interventions for many years, but this does limit the generalisability of findings to other contexts. Generalisability of the efficacy of the intervention would likely be variable across different settings. Lastly it is important to note the extensive HPS design process and the underlying formative work. Good contextual understanding is critical for intervention success, which must be culturally acceptable, locally integrated and must consider contextual baseline demographic and WASH characteristics and health status which vary significantly.

## Conclusion

The CAMPI trial evaluated feasibility of a BabyWASH HPS and recommendations to progress to a full-scale RCT in a rural, subsistence agriculture setting in Ethiopia. Not all *a priori* criteria

were met. However, overall the HPS showed mixed engagement and adherence, good acceptability and many reported secondary benefits. A larger trial with longer follow-up is feasible to implement and should assess infant health outcomes as primary endpoints. This would help determine a HPS as a viable option to reduce direct faecal-oral transmission and infant infection in this and other similar settings. Addressing identified barriers and promoting enabling factors would be necessary changes and would likely improve adherence and use.

## Supporting information

**S1 PRISMA Checklist. A checklist signposting where key information on the study design is reported in the manuscript.**
(DOCX)

**S1 Text. Modified Barrier Analysis methodology.** A description of the modified barrier analysis methodology including questions used to assess behavioural barriers to appropriate use and maintenance of the BabyWASH household playspace.
(DOCX)

**S1 Fig. The prototype design of the BabyWASH household playspace used in the CAMPI trial.** A figure containing four photographs showing the key design features of the BabyWASH household playspace.
(TIF)

**S1 Data. Supplementary data tables.** A file containing supplementary data tables including: **Table A. Playspace use behaviours and infant hygiene and playspace cleaning practices included as part of composite variables 'Appropriate use' and 'Appropriate cleaning'.** A table containing the full set of behavioural results on appropriate use and appropriate cleaning at two week, four week and both time points. **Table B. Reported daily activities and reported use or non-use of the playspace during the past 24 hours, across daily periods and study time points.** A table containing the full set of results on daily activities that BabyWASH playspace users engaged with whilst using the playspace at two week and four week time points. **Table C. Reported playspace use and non-use during daily activities in the past 24 hours across daily time periods: at two and four weeks.** A table containing results for reported use and non-use of the BabyWASH playspace during morning, afternoon and evenings. **Table D. Modified Barrier Analysis results among the study intervention group.** Two tables containing results from the modified barrier analysis covering key behavioural determinants, including perceived positive consequences, perceived self-efficacy, access and perceived social norms. **Table E. Number of samples positive for presumptive Campylobacter spp. under each category of colony count.** A table containing results for presumptive Campylobacter spp. colony count within intervention and control arm at baseline, two weeks and four weeks. **Table F. Feeding of fresh or reheated foods prepared as recommended, across study groups and time points.** A table containing results for reported feeding of fresh and reheated food at different meal times within intervention and control arm at baseline, two weeks and four weeks. **Table G. Number of meals safely prepared across time points.** A table containing results on number of safely prepared meals across intervention and control arm at baseline, two weeks and four weeks.
(DOCX)

## Acknowledgments

This study would not have been possible without the tireless efforts of the data collection teams, including Mesfine Melese, Metsinanat Eyoel, Wonsha Bulbula, Dawit Daniso, Deginet

Aklilu and Biruk Solomon. Particular special thanks go to Etsegenet Yisak Debela and Abezash Asefa Wotasa as laboratory assistants at Hawassa University College of Medicine and Health Sciences who dedicated their energy to the study. We are grateful to Afework Abraham, Frezer Girma and Endale Eyob at People In Need who kept the teams running daily. Thanks also go to the Health Extension Workers who supported the data collection teams. Finally, we thank all of the study participants who gave their valuable time and input throughout the trial duration.

## Author Contributions

**Conceptualization:** Sophie Budge, Paul Hutchings, Alison Parker, Camila Garbutt.

**Data curation:** Sophie Budge, Fitsume Woldemedhin, Mohammed Yasin Jemal, Mathewos Moges, Siraj Hussen, Hunachew Beyene.

**Formal analysis:** Sophie Budge, Sam Norton, Mathewos Moges, Siraj Hussen, Hunachew Beyene.

**Funding acquisition:** Paul Hutchings, Alison Parker, Camila Garbutt.

**Investigation:** Fitsume Woldemedhin, Mohammed Yasin Jemal.

**Methodology:** Sophie Budge, Paul Hutchings, Alison Parker, Sean Tyrrel, Camila Garbutt, Mathewos Moges.

**Project administration:** Fitsume Woldemedhin, Mohammed Yasin Jemal.

**Supervision:** Paul Hutchings, Alison Parker, Sean Tyrrel, Camila Garbutt.

**Validation:** Sam Norton.

**Writing – original draft:** Sophie Budge.

**Writing – review & editing:** Paul Hutchings, Alison Parker, Sean Tyrrel, Sam Norton, Camila Garbutt, Fitsume Woldemedhin, Mohammed Yasin Jemal, Mathewos Moges, Siraj Hussen, Hunachew Beyene.

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
