## [Decision Letter · Decision Letter 0]

23 Nov 2020

Dear Dr Hutchings,

Thank you very much for submitting your manuscript "A randomised controlled feasibility trial of a BabyWASH household playspace: The CAMPI study" for consideration at PLOS Neglected Tropical Diseases. As with all papers reviewed by the journal, your manuscript was reviewed by members of the editorial board and by several independent reviewers. In light of the reviews (below this email), we would like to invite the resubmission of a significantly-revised version that takes into account the reviewers' comments. 

We cannot make any decision about publication until we have seen the revised manuscript and your response to the reviewers' comments. Your revised manuscript is also likely to be sent to reviewers for further evaluation.

Sincerely,

Sitara SR Ajjampur

Guest Editor

Adriano Casulli

Deputy Editor

Reviewer's Responses to Questions

**Key Review Criteria Required for Acceptance?**

**Methods**

-Are the objectives of the study clearly articulated with a clear testable hypothesis stated?

-Is the study design appropriate to address the stated objectives?

-Is the population clearly described and appropriate for the hypothesis being tested?

-Is the sample size sufficient to ensure adequate power to address the hypothesis being tested?

-Were correct statistical analysis used to support conclusions?

-Are there concerns about ethical or regulatory requirements being met?

Reviewer #1: No problem

Reviewer #2: - In the abstract it is stated: "households were identified and assessed for

eligibility. Recruited households (N=100) were randomised (blinded) to intervention or

control (both n=50). " This can create a missunderstood. Was the intervention blinded? This kind of intervention is difficult to blind. 

- WASH are relevant determinants of water-related pathogens. I suggest to state what WASH data was collected. Was it infrastructure , access or use? Was it reported or observated?

- Although the intervention is not very clear in the introduction or the abstract, it is well described in methods. 

- Diarrhoea data collection might be limited by recall. Many studies observed that it could also be biased towards more cases for the ones without the intervention in order to claim for insfrastructure improvement. 

- I suggest to specify faeces collection times in methods. 

- They used observation endpoints such as "the proportion of infants in the HPS at random check", it is a good proxy.

Reviewer #3: The objectives have been stated as aims and the author can be advised to revisit it.

The design though is appropriate for the kind of trial.

The population had been described substantively, though no hypothesis had been stated but a comment had been made for reconsideration.

There are ethical concerns and the study is ethical as consent had been factored and ethical permission granted

**Results**

-Does the analysis presented match the analysis plan?

-Are the results clearly and completely presented?

-Are the figures (Tables, Images) of sufficient quality for clarity?

Reviewer #1: Does the analysis presented match the analysis plan?

-Are the results clearly and completely presented?

The sum of the available results makes it difficult to get to the point

-Are the figures (Tables, Images) of sufficient quality for clarity?

Could be improved

Reviewer #2: In overall they are ok, but perdonally I suggest to describe water and sanitation outcomes since it is a relevant determinant of health and diarrhoea.

Reviewer #3: No hypothesis presented, though the analysis plan is substantive in presenting on essential variables.

the results is well presented. 

No image but the tables and figure are okay.

**Conclusions**

-Are the conclusions supported by the data presented?

-Are the limitations of analysis clearly described?

-Do the authors discuss how these data can be helpful to advance our understanding of the topic under study?

-Is public health relevance addressed?

Reviewer #1: -Are the conclusions supported by the data presented?

Yes

Are the limitations of the analysis clearly described?

Yes

-Do the authors discuss how these data can be useful in advancing our understanding of the topic under study?

Because this is a pilot study, the authors recommend that we continue to

-Is public health relevance addressed?

Yes

Reviewer #2: The a priori expected criteria was not met. They carefuly interpret they results.

Reviewer #3: Data supported the conclusion in some ways but the statement itself did not clearly elucidate this. Thus I have commented for the concluding statement to be revisited.

Authors did made remarks on the positive observations from the intervention, however how data can be helpful is not explicitly discussed

**Editorial and Data Presentation Modifications?**

Reviewer #1: Please see the attachment

Reviewer #3: Minor Revision.

**Summary and General Comments**

Reviewer #1: Interesting submission which is almost too voluminous in terms of information and results. 2 complementary papers highlighting the essential points would be interesting. but it is only a proposal.

Reviewer #2: (No Response)

Reviewer #3: Depending on the context, the above trial can be foundation for the need to address child welfare and growth monitoring in communities like in Ethiopia, however generalisation of the efficacy of the intervention other setting may significantly differ including also if the study lasted much longer in the same setting.

PLOS authors have the option to publish the peer review history of their article (what does this mean?). If published, this will include your full peer review and any attached files.

Reviewer #1: No

Reviewer #2: No

Reviewer #3: Yes: Yaya Camara
---

## [Decision Letter · Decision Letter 1]

12 Apr 2021

Dear Dr Hutchings,

Thank you very much for submitting your manuscript "A randomised controlled feasibility trial of a BabyWASH household playspace: The CAMPI study" for consideration at PLOS Neglected Tropical Diseases. As with all papers reviewed by the journal, your manuscript was reviewed by members of the editorial board and by several independent reviewers. The reviewers appreciated the attention to an important topic. Based on the reviews, we are likely to accept this manuscript for publication, providing that you modify the manuscript according to the review recommendations. 

Sincerely,

Adriano Casulli, PhD

Deputy Editor

Reviewer's Responses to Questions

**Key Review Criteria Required for Acceptance?**

**Methods**

-Are the objectives of the study clearly articulated with a clear testable hypothesis stated?

-Is the study design appropriate to address the stated objectives?

-Is the population clearly described and appropriate for the hypothesis being tested?

-Is the sample size sufficient to ensure adequate power to address the hypothesis being tested?

-Were correct statistical analysis used to support conclusions?

-Are there concerns about ethical or regulatory requirements being met?

Reviewer #1: (No Response)

Reviewer #2: The authors adressed all comments and suggestions.

Reviewer #4: Yes with a few questions ...

In the Aims section, the authors write, “As a feasibility trial, a sample size calculation was not performed. A target of 100 households was deemed sufficient to inform researchers about practicalities of running the trial and for sufficient precision to estimate rates of recruitment, retention, and trial outcomes.” What criteria did they use to determine 100 is “sufficient”?

In the statistical analysis section, the authors write, “Trial outcomes are displayed with 95% confidence intervals (CI).” Can the authors please clarify here which confidence intervals (e.g., approximate, exact) were calculated and reported?

In the statistical analysis section, the authors write, “A Generalised Estimating Equation (GEE) was used as a semiparametric model, using a robust variance estimator and an unstructured working correlation matrix. A binary logistic GEE estimated factors associated with ‘Appropriate use’ and ‘Appropriate cleaning’ at two and four weeks. Models were initially run separately: however the merged composite variable of ‘Appropriate use and cleaning’ showed no difference in parameter estimates between models and is presented.” Was the study powered to conduct this analysis and achieve significant differences or is this being done for preliminary/exploratory purposes only? The authors do comment on the limitations in the discussion so it may be helpful to emphasize this when presenting the results.

In the statistical analysis section, the authors write, “Results are expressed as populated averaged odds ratios (ORs) with 95% CI.” Can they please clarify which confidence intervals were reported here? The same comment for the statement, “Change in diarrhoeal and Campylobacter prevalence between study groups was estimated using a GEE intercept-only model with OR and 95% CI.” Table 4 is the only place where Wald is specified, so clarification would be helpful given the small sample size.

**Results**

-Does the analysis presented match the analysis plan?

-Are the results clearly and completely presented?

-Are the figures (Tables, Images) of sufficient quality for clarity?

Reviewer #1: (No Response)

Reviewer #2: The authors adressed all comments and suggestions from previous review.

Reviewer #4: Yes

**Conclusions**

-Are the conclusions supported by the data presented?

-Are the limitations of analysis clearly described?

-Do the authors discuss how these data can be helpful to advance our understanding of the topic under study?

-Is public health relevance addressed?

Reviewer #1: (No Response)

Reviewer #2: The authors adressed all comments and suggestions from previous review.

Reviewer #4: Yes

**Editorial and Data Presentation Modifications?**

Reviewer #1: (No Response)

Reviewer #2: (No Response)

Reviewer #4: (No Response)

**Summary and General Comments**

Reviewer #1: (No Response)

Reviewer #2: The trial limitations have been mentioned and discussed. It is a matter of the journal decide if this work should be published in their journal.

Reviewer #4: The paper is well written and the analyses (although limited) are appropriate. The authors have adequately addressed the reviewers' comments/concerns and improved the manuscript through this revision. I included a few minor comments/questions above.

PLOS authors have the option to publish the peer review history of their article (what does this mean?). If published, this will include your full peer review and any attached files.

Reviewer #1: No

Reviewer #2: No

Reviewer #4: No

Figure Files:

Data Requirements:

Reproducibility:

References

---

## [Editor Report · Decision Letter 2]

28 May 2021

Dear Dr Hutchings,

We are pleased to inform you that your manuscript 'A randomised controlled feasibility trial of a BabyWASH household playspace: The CAMPI study' has been provisionally accepted for publication in PLOS Neglected Tropical Diseases.

Best regards,

Sitara SR Ajjampur

Associate Editor

Adriano Casulli

Deputy Editor

---

## [Editor Report · Acceptance letter]

5 Jul 2021

Dear Dr Hutchings,

We are delighted to inform you that your manuscript, "A randomised controlled feasibility trial of a BabyWASH household playspace: The CAMPI study," has been formally accepted for publication in PLOS Neglected Tropical Diseases.

Best regards,

Shaden Kamhawi

co-Editor-in-Chief

Paul Brindley

co-Editor-in-Chief
